# Towards Scalable Imitation Learning for Multi-Agent Systems with Graph Neural Networks

## Abstract

We propose an implementation of GNN that predicts and imitates the motion behaviors from observed swarm trajectory data. The network's ability to capture interaction dynamics in swarms is demonstrated through transfer learning. We finally discuss the inherent availability and challenges in the scalability of GNN, and proposed a method to improve it with layer-wise tuning and mixing of data enabled by padding.

## 1 Introduction

In multi-agent systems (MAS) (Arai et al., 2002), a group of agents has to effectively collaborate in order to solve a complex task in a parallel manner. Individual agents have to synchronize their actions and behaviors so as to produce a whole that is more than the sum of its parts. This synchronization is a core characteristics of MAS and is of vital importance to a wide range of modern application domains such as smart agriculture, military demining, or warehouse robotics. However, developing control strategies that allows a group of agents to perform a task jointly is still considered to be a complex challenge. This statement is particularly true for mobile agents that need to dynamically act in a physical space. To overcome this challenge, a variety of formalisms have been proposed to support the modeling and design of multi-agent and multi-robot systems. A prominent approach, called *Boids* or bird-oid, was proposed in Reynolds (1987). In the Boids model, the system behavior emerges from the interplay of a few simple building-blocks. While simple to implement, it can be challenging to control the overall behavior of the swarm due to the emergent nature of the system dynamics. Other important formalisms for the specification of MAS are based on graph theory (Mesbahi & Egerstedt, 2010), game theory, and formal languages. However, the latter approaches still require substantial modeling, coding, and verification efforts in order to yield practical controllers that can scale with the size of the system, i.e., the number of agents. Another obvious choice to reduce or eliminate modeling effort is to use machine learning. Reinforcement learning (Sartoretti et al., 2019), in particular, has previously gained attention for synthesizing collective behavior in an MAS. However, due to the curse of dimensionality, reinforcement learning approaches often struggles to scale to systems with large number of agents.

In this paper, we present an approach for learning multi-agent coordination through imitation (Schaal, 1999). Rather than specifying the system dynamics, the designer only has to provide demonstrations of successful coordination behavior. In turn, this data set is used to learn a neural network representation of the underlying dynamics and rules of interaction. More specifically, we leverage recent insights regarding graph neural networks (GNN) (Battaglia et al., 2018) to learn a structured model of the dynamics. In contrast to traditional neural networks which take a vector, matrix or tensor as input, our graph neural network processes a graph representation of the multi-agent system. An implementation example of GNN can be seen in Kipf et al. (2018), where the ability of GNN to better predict the motion of a simple physically interacting particles is highlighted. We show that the specific GNN implemented in this paper can accurately capture the dynamics of more complex multi-agent motion behaviors given a set of demonstrations. We also discuss and analyze difficulties in scaling learned models up to an MAS with a larger set of agents. Based on this discussion, we then propose a refinement training procedure to address these issues. The refinement procedure helps tuning a learned model to a system with a different number of agents, i.e., scalability along the size of the MAS.

## 2    MODEL

### 2.1    OVERVIEW

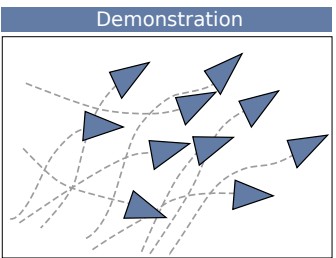 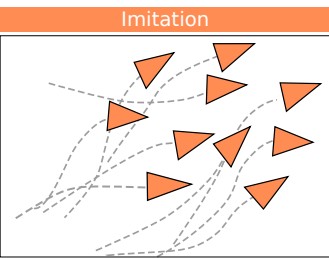 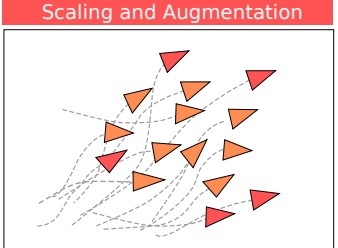

Figure 1: A depiction of the training and inference. For training the trajectories of a set of agents is recorded as demonstrations. After we train our GNN model, it can be used to create a copy of the swarm. In addition, it is possible to create larger and smaller swarms by changing the number of agents.

In this section, we describe our methodology for imitation learning of MAS behavior. Fig. 1 depicts an overview of both the training and inference process for MAS[1]. The input to the learning process is a set of demonstration trajectories, or execution traces, of the agents in the swarm. Execution traces are discretely sampled trajectories specifying the position and velocity of each agent at time step $t$. These trajectories are used to train a graph neural network to model the group behavior observed in the swarm. Our network takes the time series as input and generates the predicted next steps in the time series. The prediction error is measured by the mean squared error between output states and expected states from the true time series. The task of our network is to model the motion dynamics of the MAS with minimum prediction error. Without the assumption that the dynamics is first-order Markovian, states earlier than the present state carry predictive power for future states and must be accounted for. Historical states and their higher order relations with the current states, for example, velocity and acceleration, are necessary for making accurate predictions. Therefore, we apply a series of one-dimensional convolutional layers to process a history states for the agents. The 1D convolution acts along the time steps of each agent and abstracts a representation of its past as the starting point for prediction. This representation will then be taken as the input of the GNN.

The interaction of the swarm system being embedded in a graph, the GNN directly models the information propagation within the graph by explicitly computing the interaction between nodes and account for the influence from the interactions to each node to update the graph's state. The procedures can in general be identified as 3 steps. Firstly, interactions between all pairs of nodes connected through an edge are computed by a function universal across all edges. The interaction, often called edge state, is determined only by its current state and the two nodes at both ends. Secondly, centered on each individual node, all interactions directed to it are aggregated by another function universal to all nodes to form the local influence to the node. This step is sometimes called edge aggregation. Lastly, a third function universal to all nodes updates each node state based on the node's current state and aggregated influence, thus completing the state update of the entire network. Evolution of graph dynamics is fully described by applying the procedures repeatedly. Details of structure and training procedures will be explained in the following subsections.

### 2.2    DATA STRUCTURE

The input time series of motion states from an instance of swarm simulation are arranged to have the shape of $T \times N \times D$, where $T$ is the number of time steps, $N$ is the number of agents, and $D$ is the dimension of the state vector $s_i(t)$ of agent $i$ at step $t$, where $i, t \in \mathcal{N}, i < N, t < T$. The state vector contains both the position and velocity, i.e., $s_i(t) = [\mathbf{x}_i(t), \dot{\mathbf{x}}_i(t)]$, where the square brackets stand for concatenation of the position $\mathbf{x}_i(t)$ and velocity $\dot{\mathbf{x}}_i(t)$. In all subsequent experiments presented in this paper, all agents live in a two-dimensional space, which lets the dimension of state vectors $\mathcal{D} = 4$. A sequence with window length $T_w$ of history states $s_i(t - T_w + 1), s_i(t - T_w +$

---

[1]We will henceforth use the term "swarm" and MAS interchangeably.

2), ..., $s_i(t-1)$, $s_i(t)$ are processed to output the next future state $s_i(t+1)$. Long term prediction is realized by repeatedly appending the predicted new step to the sequence of history state, dropping the earlier states and feeding the new sequence of length $T_w$ as input.

## 2.3    1D CONVOLUTIONS

The goal of one-dimensional convolutions is to abstract a concise representation of the historical states. The 1D convolutions with no padding condense the time sequences for each agent, until the output after the last 1D convolution layer is reduced to a $1 \times N \times C$ shaped tensor, with $C$ being the number of channels of the last layer. For history window length $T_w$ and a universal kernel size of $K$, this would require at least $\lceil (T-1)/(K-1) \rceil$ layers since each layer condenses the length by $K-1$. We typically choose the number of 1D convolution layers $L$ and $T_w$ such that they satisfy $T_w = L(K-1) + 1$. $K = 3$ is what we have used throughout the experiments. As the 1D convolutional filters slide through the entire time series of trajectories of length $T$, $T_s = T + 1 - T_w$ sub-sequences are condensed, each serving as the basis for prediction, and are sent to the following Graph Neural Network module to produce the predicted states for their corresponding next steps.

## 2.4    GRAPH NEURAL NETWORK

MAS along with their interactions are embedded in a directed graph $\mathcal{G} = (\mathcal{N}, \mathcal{E})$, where $\mathcal{N}$ is the set of nodes which represent the agents, and $\mathcal{E} = \{(i,j)|i,j \in \mathcal{N}, i \neq j\}$ is the set of edges whose element $(i,j)$ represents the influence from agent $i$ to agent $j$ through interaction. The state of the node $i \in \mathcal{N}$ is a vector of dimension $d_v$, as $\boldsymbol{v}_i \in \mathbb{R}^{d_v}$, while the state of an edge $(i,j), i,j \in \mathcal{N}$ $i \neq j$ is a vector of dimension $d_e$, that is, $\boldsymbol{e}_{ij} \in \mathbb{R}^{d_e}$. In swarm motion, such interactions include pulling, pushing and steering, and the edge states may be interpreted as an abstraction of the physical forces.

For each of the $T_s$ slices from the output of 1D convolution module, the GNN module takes the $N \times C$ tensor as states of the $N$ nodes. The node state of the next step inevitably is influenced by the interactions the node experiences. The interactions depend on the current and previous states of the involved nodes. As introduced earlier, this information passing process is realized by GNN, formulated by 3 functions below:

$$\boldsymbol{e}_{ij} = \phi^e(\boldsymbol{v}_i, \boldsymbol{v}_j), \tag{1}$$

$$\bar{\boldsymbol{e}}_i = \psi^{\bar{e}}(\sum_{j \in \mathcal{N}_i} \boldsymbol{e}_{ji}), \tag{2}$$

$$\boldsymbol{v}'_i = \phi^v(\boldsymbol{v}_i, \bar{\boldsymbol{e}}_i), \tag{3}$$

where $\mathcal{N}_i$ is the set of source nodes from the edges directed to node $i$. Function $\phi^e$, computes the influence from node $i$ to node $j$. Function $\psi^{\bar{e}}$ aggregates the influences from all sources for node $i$ as the total influence $\bar{e}_i$. And finally, function $\phi^v$ transforms the total influence and the node's historical states into the prediction of the next step. Definition similar to these equations and thorough discussion of graph neural network may be found in Battaglia et al. (2018). All three functions are each approximated by a multi-layer perceptron (MLP).

Note that in Eq. 2, summation on all edge state vectors is taken before $\psi^{\bar{e}}$ is applied. It can be argued that influences from different sources may need to be weighted. However, the distinction between edges relies on the distinction of the nodes attached, which is already taken care of by function $\phi^e$. This also brings forth the point, that if the distinction between edges is known before hand, a set of individual functions $\phi^e_k(\mathbf{v}_i, \mathbf{v}_j)$ can be created to cover the different interactions, where $k \in T_E$ is the label of edge $ij$ among the set of known types $T_E$. The interaction influence between nodes is computed by $\phi^e_k(\mathbf{v}_i, \mathbf{v}_j)$ with respect to the edge labels $k$ as part of input.

In principle, the group of GNN operations may be stacked one after another just like layers of multi-layer perceptron, where the output vector $\boldsymbol{v}'_i$ of intermediate layers would serve as hidden states and only the output of the last layer is considered the update for the next step. In that case, interaction horizon gets expanded from the nearest neighbors, since information propagates further down the edges with each GNN layer added. Nevertheless, in this paper, we only employ one GNN layer in our model, as is appropriate for the physical settings of a swarm system.

It is worth pointing out that $\mathbf{v}i$ and $\mathbf{v}'_i$, despite both being the "node state" before and after GNN operations, may carry different meanings semantically, as in the practice of this paper where we treat

the output $\mathbf{v}'_i$ as the predicted change between the current state vector $\mathbf{s}_i(t)$ and future state vector $\mathbf{s}_i(t+1)$ of agent $i$, $\mathbf{s}_i(t+1) = \mathbf{v}' + \mathbf{s}_i(t)$, while we assign the representation of history states to $\mathbf{v}_i$. The core algorithm of our GNN is presented in Algorithm 1. Parallelization in the actual implementation is not included.

---

**Data:** History states $\mathbf{S}_{(1:T_w,1:N,1:D)}$, Next state $\boldsymbol{S}^*_{(1:N,1:D)}$, Edge type one-hot labels
$\qquad \mathbf{E}_{(1:N,1:N,1:T_E)}$
**Result:** Conv1D parameters $w$, Function $\phi^e, \psi^{\bar{e}}, \phi^v$ parameters $\theta_1, \theta_2, \theta_3$
Random initialize $w, \theta_1, \theta_2, \theta_3$ $\boldsymbol{S}'_{(1:N,1:D)} \leftarrow \mathbf{S}_{(T_w,1:N,1:D)}$ $\qquad\qquad$ // Current state
$\boldsymbol{V}_{(1:N,1:D)} \leftarrow \text{Conv1D}(\mathbf{S}_{(1:T_w,1:N,1:D;w)})$ **for** $i$ *in* $1:N$ **do**
$\quad$ | $\quad \boldsymbol{v} \leftarrow \boldsymbol{V}_{(i,1:D)}$ $\bar{\boldsymbol{e}} \leftarrow \mathbf{0}$
$\quad$ | $\quad$ **for** $j$ *in* $1:N$ **do**
$\quad$ | $\quad$ | $\quad \boldsymbol{u} \leftarrow \boldsymbol{V}_{(j,1:D)}$ **for** $k$ *in* $1:T_E$ **do**
$\quad$ | $\quad$ | $\quad$ | $\quad \boldsymbol{e} \leftarrow \phi^e(\boldsymbol{v}, \boldsymbol{u}) * \mathbf{E}_{(i,j,k)}$
$\quad$ | $\quad$ | $\quad$ | $\quad \bar{\boldsymbol{e}} \leftarrow \bar{\boldsymbol{e}} + \boldsymbol{e}$
$\quad$ | $\quad$ | $\quad$ **end**
$\quad$ | $\quad$ | $\quad \bar{\boldsymbol{e}} \leftarrow \psi^{\bar{e}}(\bar{\boldsymbol{e}})$
$\quad$ | $\quad$ **end**
$\quad$ | $\quad \boldsymbol{v}' \leftarrow \phi^v(\boldsymbol{v}, \bar{\boldsymbol{e}})$
$\quad$ | $\quad \boldsymbol{S}'_{(i,1:D)} \leftarrow \boldsymbol{S}'_{(i,1:D)} + \boldsymbol{v}'$
**end**
$L \leftarrow \text{MSE}(\boldsymbol{S}'_{(1:N,1:D)}, \boldsymbol{S}^*_{(1:N,1:D)})$
$(w, \theta_1, \theta_2, \theta_3) \leftarrow \text{Adam}(L, (w, \theta_1, \theta_2, \theta_3))$

**Algorithm 1:** A training step on swarm dataset

---

### 2.5 Loss Function and Training

Prediction error is measured as the mean squared error (MSE) between the predicted states $\mathbf{s}_i(t+1)$ and the ground truth $\mathbf{s}^*_i(t+1)$ from motion data. Supervised training is performed to adjust the parameters in the functions by minimizing the MSE between $\mathbf{s}_i(t+1)$ and $\mathbf{s}^*_i(t+1)$ over all agents and all time steps.

Given that the simulated motion data is discretely sampled from trajectories, how fine grained the sample rate is affects the manifest of the error. Because of this, the error is normalized by the average "natural skip" in the ground truth data, i.e., the MSE of state vectors between two consecutive steps in the ground truth trajectories, $\bar{L}$,

$$\bar{L} = \frac{1}{2DN(T-1)} \sum_{t=1}^{T-1} \sum_{i=1}^{N} (\boldsymbol{s}^*_i(t+1) - \boldsymbol{s}^*_i(t))^2 \qquad (4)$$

Normalized error $L_{norm} = \frac{L}{\bar{L}}$ rectifies the dependence of prediction error on the scale of the intrinsic spacing in the ground-truth data.

Multistep prediction is enabled by appending the predicted state $\mathbf{s}_i(t+1)$ back to the sequence of $T_w$ states and taking the new last $T_w$ states as the history ground for future prediction. Multi-step prediction builds up predictions in the far future on the predictions in the near future, and thus imposes a higher demand on the accuracy and encourages the model to learn the true mechanisms better, despite harder to train. We gradually increase the difficulty of the prediction task by increasing the required number of prediction steps. This curriculum learning (Bengio et al. (2009)) starting with short term prediction helps to guide the model faster in the earlier stage while still granting good long term prediction power later.

## 3 Scalability

Keen observation on GNN's update rules Eq. 1-3 reveals that all 3 equations operate locally on the neighborhood of nodes. In other words, global attributes such as the number of nodes are not involved. So long as swarm systems share the same set of mechanisms, manifested by the same set of

functions $\phi^e$, $\psi^{\bar{e}}$ and $\phi^v$ in their underlying interaction graphs, the same GNN should be applicable to all the systems without the need for modification to the parameters. Although we argue later that Eq. 2 and 3 are not directly transferable, the statement still stands true that no change to the structure of the GNN is required.

Indeed, closer inspection of Eq. 2 raises suspicion that the number of edges connected to a node $i$ may be implicitly correlated with the number of nodes in the graph. In reality, it is reasonable to expect agents to interact with more neighbors in a larger swarm. Despite that the swarms share the same set of functions, when transferring to a swarm of a different size, function $\psi^{\bar{e}}$ may be exposed to an unseen range simply due to more edges to be aggregated. Adding to the fact that MLPs are poor at extrapolation (Hettiarachchi et al., 2005), function $\psi^{\bar{e}}$ may fail to respond correctly, leading to a drop in accuracy of the GNN.

The scalability issue caused by poor extrapolation ability by MLP of Eq. 2 and Eq. 3 would become prominent in more realistic swarm models. In realistic swarms, not only does the size of agent neighborhood change with the size of the swarm, but the neighborhood may change dynamically as well. When long interaction range is assumed, agents are often effectively fully connected, extending the local neighborhood to the entire connectivity graph. The strong correlation between local connectivity and swarm size under this assumption makes trans-swarm application of a trained GNN almost impossible.

It is plausible that the simple summation in Eq. 1 is a bad choice. However, one may argue that popular numerical models of swarms such as Boid (Reynolds, 1999) and Vicsek (Helbing et al., 2000) that achieve high level of authenticity are partly or strictly physics based and are built on superposition of accelerations caused by pair wise interactions, realized by simple summation. In the scope of this paper, without deviating away from the proposed update rules of GNN, we propose a layer-wise tuning method realized by data padding as an attempt to reduce GNN's difficulty to scale.

An intuitive answer to treat the lack of responsiveness is to adjust the functions for edge aggregation and node update through transfer learning. To avoid catastrophic forgetting (Kirkpatrick et al., 2017), however, the model has to constantly revisit previous data set. Although GNN in principle can freely adapt to different numbers of nodes, having a mixed dimension of input data poses problems to tensor based training frameworks. We note that a node with no interaction with any other node in a graph exerts no influence on or be affected by the rest of the graph, and in reverse, adding an isolated node to a graph does not change the dynamics of the members of the original graph. This immediately implies that the input data of a certain swarm can be padded with an arbitrary number "ghost" agents with 0 edges in the connectivity matrix. Through state update of the graph, the edge message between a "ghost" and any other node is never accounted for, so the GNN works equally well on the real agents. In this way, input data may have a uniform shape in the dimension of $N$ when mixed with swarms of different sizes. We choose to initialize the states of "ghost" with zeros. Note that history states $\mathbf{S}_{(1:T_w,1:N,1:D)}$ in Algorithm 1 is padded to the maximum $N$ in dataset with mixed $N$'s. During transfer learning, $w, \theta_1, \theta_2$ are locked, and only $\theta_3$ is to be updated in the last line.

## 4 EXPERIMENTS

Before exploring the solutions to scalability, we first show that our model performs better than other models in motion prediction on swarms of the size it is trained on (Table 1). We test the accuracy of models' prediction on simulated Boid data and Vicsek data. In a Boid system, agents converge towards a goal while trying to match neighbors' position and velocity without collision. In a Vicsek system, agents only avoid each other via repulsive force. We artificially inject goals to Vicsek systems similar to those in Boid systems to guide the agents. All three models are trained on both the Boid and the Vicsek data, with 10 agents only for each model, but are tested on respective models of different sizes. Table 1 shows the MSE on test set for long term prediction upto 40 steps. The table also illustrate how non-GNN base model such as Seq2Seq (Sutskever et al., 2014) is not able to be applied to swarms of different sizes. Kipf's GNN (Kipf et al., 2018) being tasked with inferring the latent edge types, comes with the burden of a more complex structure, and is hard to train. Our GNN variant, however, has the advantage of knowing the type labels of edges and a cleaner structure, can more easily capture the dynamics of the interactions, and produce accurate long term predictions.

Table 1: Model comparison for prediction error

| MODEL | LOSS | | | | | |
|---|---|---|---|---|---|---|
| | 5 boids | 10 boids | 20 boids | 5 vicseks | 10 vicseks | 20 vicseks |
| Seq2Seq | - | 1.45±0.14 | - | - | 1.50±0.13 | - |
| Kipf's GNN | 111±16 | 11.7±0.88 | $\sim 10^4$ | $\sim 10^4$ | 10.6±0.6 | $\sim 10^6$ |
| Our GNN | 0.93±0.09 | 0.41±0.14 | 1.69±0.15 | 0.24±0.02 | 0.05±0.03 | 0.32±0.04 |

Not only that, it already shows better scalability on data of unseen sizes, possibly due to its slimmer structure. Samples of predicted long term trajectories are shown in the left pane of Fig. 2.

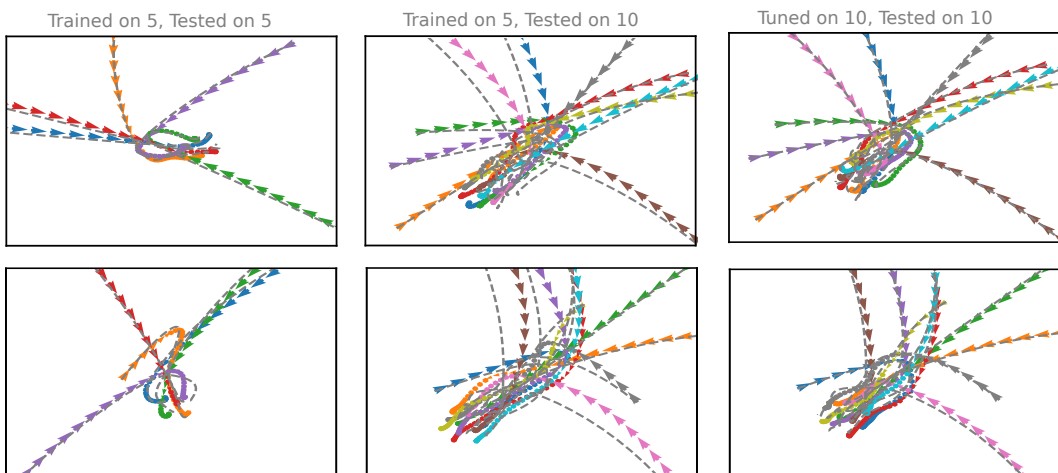

Figure 2: Predicted trajectories (colored arrows) overlaid on ground truth (gray dash lines). Only $T_w$ initial states are supplied, and the prediction is done repeatedly upto 40 steps in future. *Left*: Trajectories of 5 boids by a network trained with data of 5 boids. *Middle*: Trajectories of 10 boids by the same network used in the left pane. *Right*: Trajectories of 10 boids by the network after tuning on padded data of 5 and 10 boids.

Secondly, following earlier discussion in Section 3 to prove the hypothesis that GNN naturally scales with $N$, we designed a simplistic swarm model called *chaser* to run experiments on as follows. A chaser is an agent that chases another particle with acceleration proportional to its displacement from its target. In a chaser swarm, each particle is only influenced by one other particle as its target, which means the interaction graph has only one edge attached towards each agent. The summation in Eq. 2 becomes trivial as it equals the single edge state associated with agent $i$, and function $\psi^{\bar{e}}$ can be absorbed into function $\phi^e$ when chained together. The second input to Eq. 3,which is the aggregated edge message $\bar{e}$, directly inherits the single edge message. So, just like each chaser is oblivious to the rest of the group except for its target, the update rules are independent of $N$.

We trained our GNN on the simulated motion data from a chaser swarm of only a minimal number of agent, $N = 3$. Without any modification to the trained GNN, the test results on chaser swarms of various sizes $N = 3, 5, 10, 20, 100$ show perfect replication of the ground truth trajectory (Fig. 3), given only one window length of $T_w$ initial steps as starting points. This simplistic case also seems to indicate that a learned function $\phi^e$ that governs the universal pair-wise interactions is transferable across swarms of the same type.

To investigate how scalability is challenged by the change in size of the neighborhoods through summation in Eq. 2, we modify the dynamic rules of the chaser model, such that each agent may now chase more than 1 target. We again train our model only on a chaser swarm with $N = 3$ and $M = 1$, $M$ being the number of targets each chaser has. The trained model is tested on various combinations of $N$ and $M$. It is immediate to notice the accuracy degradation in the first row of Table 2. To illustrate the shift in the input range to function $\psi^{\bar{e}}$ and subsequently function $\phi^v$, we

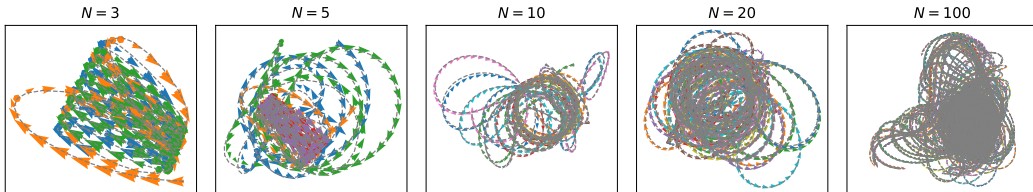

Figure 3: Predicted trajectories for chaser swarms of different sizes $N$. Colored arrows mark the predicted velocities and positions, against the grey dashed lines as the ground truth trajectories. Every agent in these systems chases only 1 other agent. Training was done only on systems with $N = 3$.

plot the distribution of the norm of output vectors of the functions' preceding components (Fig. 4). With the output of first function sharing similar distributions across systems, it is unsurprising that the ones with higher $M$'s have the distribution of summation shifted to the right. Less overlapping is shared by the distributions when the distinction in $M$ is larger, though same $M$ still guarantees almost identical distribution. The strong resemblance by the distributions of function $\phi^e$'s output favors the presumption that the learned function $\phi^e$ in GNN from one swarm is readily transferable to other swarms with shared interaction rules.

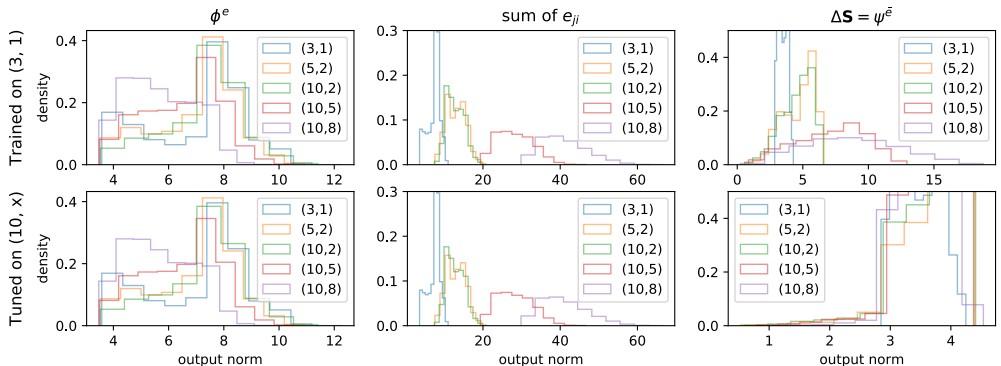

Figure 4: Distribution of output vector norms of critical components in our GNN, before and after tuning, on data sets with various $(N, M)$. *Left*: Output of function $\phi^e$. *Middle*: Output of edge message summation before entering function $\psi^{\bar{e}}$. *Right*: Final output of the network from function $\phi^v$, as changes to agent states.

Next we show in Table 2, how tuning only the Eq. 2 and Eq. 3 with padded data extend the performance on swarms of larger neighborhoods. For the chaser model, after training only on a dataset with 3 agent, each with 1 target ($N = 3, M = 1$), only the validation set with the same size of neighborhood sees the same level of performance (e.g. $N = 3, M = 1$ and $N = 5, M = 1$). We tune the model with Eq. 1 locked on a mixed set of chasers with $N = 10$, and $M$ varying between 1 and 9. Note at this moment, padding is not necessary, because the nature of chaser model also allows a variety of neighborhood sizes without changing the swarm size. After tuning, performance on $M$ less than 9 achieves similar levels, and is also greatly improved on larger swarms with larger neighborhoods. We see in Fig. 4 that tuning has corrected functions' response to unseen ranges caused by aggregation on higher number of edge messages. The apparent resemblance of the corrected distributions is due to the design of the chaser model, in which a particle chases the average position of all its targets, thus resulting in similar changes regardless of $M$. For other partly physics based models, the bias is not applicable, since having a higher number of interactions is often coupled with stronger changes. Fig. 5 shows the quality of the prediction even for larger unseen neighborhoods.

The ground for tuning only Eq. 2 and Eq. 3 is based on the premises that training on one dataset, the network has already learned Eq. 1 as interaction rules universal to all sizes of the same swarms. To show the importance of a learned function $\phi^e$ and rule out the suspicion that the performance is solely determined by subsequent functions, we lock function $\phi^e$ right after random initialization and

Table 2: Chaser evaluation before and after tuning

| TRAIN | VALIDATION SET | | | | | | | | |
|---|---|---|---|---|---|---|---|---|---|
| $(N, M)$ | (3,1) | (5,1) | (5,2) | (10,2) | (10,5) | (10,8) | (20,9) | (20,12) | (20,17) |
| (3, 1) | 0.051 | 0.036 | 17.02 | 14.27 | 104.93 | 275.3 | 350.7 | 701.4 | 1932 |
| (10,x) | 0.381 | 0.223 | 0.0939 | 0.0799 | 0.142 | 0.278 | 0.291 | 0.608 | 1.424 |

Figure 5: Predicted trajectories after tuning against the ground truth for chaser model with various swarm sizes $N$ and neighborhood sizes $M$. Only the trajectories of 2 agents are shown for legibility.

train the other functions from scratch, the training error is never able to be brought down below 1 even for 1 step prediction, let alone long term prediction.

For Boid, we assume the agents are fully connected. The number of interaction edges, i.e. the size of neighborhood, is equivalent to the size of the swarms. We initially train the network from scratch on 5 boids and tune it with parameters of Eq. 1 locked on a mix of padded data of 5 and 10 boids (Table 3). Compared to networks that is trained only on 5 boids or 10 boids from scratch, this tuning and padding method preserves good performance on both data sizes and ones in between, while also avoiding catastrophic forgetting that appears in other transfer learning processes without mixed padded data. Trajectories in Fig. 2 show the improved performance from untuned network to a tuned one. As comparison, catastrophic forget occurs when revisiting of old data is not available. For instance in Table 3, the network that is trained first with 5 boids and tuned only with 10 boids, performance peak has quickly moved from 5 boids to 10 boids. After tuning on 20 boids, the peak moved to 20 boids as well.

Table 3: Training method comparison

| METHOD | LOSS | | | | |
|---|---|---|---|---|---|
| | 3 | 5 | 7 | 10 | 20 |
| 5 | 0.58±0.06 | 0.14±0.03 | 0.48±0.08 | 1.66±0.28 | 9.6±2.0 |
| 10 | 1.96±0.45 | 0.93±0.09 | 0.50±0.11 | 0.41±0.14 | 1.69±0.15 |
| 20 | 3.01±0.403 | 1.78±0.11 | 1.31±0.06 | 0.86±0.03 | 0.64±0.01 |
| 5→10 | 9.04±8.46 | 1.50±0.85 | 0.42±0.04 | 0.29±0.02 | 1.68±0.31 |
| **5→5-10** | **0.69±0.13** | **0.13±0.02** | **0.33±0.08** | **0.28±0.02** | **1.74±0.27** |
| 10→20 | 13.1±13.7 | 6.21±6.16 | 3.41±2.90 | 1.36±0.71 | 0.62±0.03 |

## 5 CONCLUSION

In this paper, we proposed an implementation of GNN that predicts and imitates the motion behaviors from observed swarm trajectory data. The network is combined with curriculum learning to achieve high accuracy prediction for arbitrary time steps. We demonstrated the network's ability to capture interaction dynamics in swarms through transfer learning. We discussed the availability and challenges in the scalability of GNN, and proposed a method to improve it, by using layer-wise tuning and mixing of data enabled by padding.

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
