# OpenReview forum: "Towards Scalable Imitation Learning for Multi-Agent Systems with Graph Neural Networks"
_ICLR.cc/2020/Conference — Reject_

### Official Review · AnonReviewer2 · 2019-10-21
**Official Blind Review #2**

**Rating:** 3

**Review:**

Review for "Towards Scalable Imitation Learning For Multi-Agent Systems with Graph Neural Networks".

The paper proposes a new time series model for learning a sequence of graphs.

I vote to reject the paper for three reasons.

1. Lack of significance. Algorithm 1 is essentially supervised learning of an autoregressive model with a GNN. The GNN definition is expanded out in the pseudocode, but seems to be completely standard. Moreover, Section 2.5 is also completely standard - you are using a sum of standard MSE losses. It should be much more concise. Also, there is typically no point scaling the loss by D if you use Adam (because the scaling they will eventually cancel out).

2. Poor awareness of prior work. What you call "scalability issue caused by poor extrapolation" is normally called poor generalization. Improving generalization is an established problem in ML. You should cite some of that work. See [1] and follow the references.

3. Poor experimental evaluation. Your experiments show that the competing GNN method (what you call Kipf's GNN) didn't converge in certain settings. Robust GNN implementations exist that converge for a wide range of reasonable graphs. You should have used one of them.

As things stand, because the criticisms concern the core of the submission, I have doubts the quality of the paper can be improved enough within the ICLR revision time to obtain an "accept" score. However, I wanted to encourage you not to give up. The building blocks that you have in the paper are very relevant and can be a basis for impactful work. You also write well (despite some minor issues). I hope these skills will help you make great submissions in the future!

Minor points:
1. I would not call what the paper is doing "imitation learning". Imitation learning normally means you have a controllable system and want to learn a policy from expert demonstrations. What this paper does is commonly referred to as learning an autoregressive time series model.
2. Avoid colorful language. For example, the sentence: "the apparent resemblance of the corrected distributions in the final output may inspire an inductive bias as part of training objective" is unclear.

[1] https://papers.nips.cc/paper/7176-exploring-generalization-in-deep-learning.pdf

**Experience Assessment:**

I have read many papers in this area.

**Review Assessment: Checking Correctness Of Derivations And Theory:**

N/A

**Review Assessment: Checking Correctness Of Experiments:**

I assessed the sensibility of the experiments.

**Review Assessment: Thoroughness In Paper Reading:**

N/A

---

> ### Author Response · Authors · 2019-11-15
> **Responses**
>
> We agree the introduction and explanation of our model are partly based on the standard practices of GNN. However, we point out that the emphasis and significance of our paper is the scalability of GNN based networks, and the introduction of our model is to show a working model. We dedicated half our paper to the analysis on the origins of difficulties in scalability, and the exploration of solutions. We proposed a padding method that allows mixed input data of graphs with different sizes, and in turn, enables "generalization" to some extent through training on mixed data.

---

### Official Review · AnonReviewer1 · 2019-10-23
**Official Blind Review #1**

**Rating:** 1

**Review:**

This work considers sequence prediction problems in a multi-agent system (MAS), which I think is different from imitation learning problems where agents try to mimic experts’ behavior given histories or states. In that sense, I think the title of this work should be changed so that readers are not confused.

The main idea of this work is to use (1) graph neural networks (GNNs) to learn the abstract information among multiple agents, (2) use 1D convolution to extract historical features of each agent, (3) and minimize the MSE loss function between true and predicted states to make expected successor states of multiple agents more accurate. The experiments show training in a small number of agents can be generalized and transferred to the setting in which there is a large number of agents.

Although the proposed algorithm is practically useful, I believe the submission is premature to be accepted at a conference due to (1) the lack of comparison with existing works on multi-agent (reinforcement and imitation) learning and (2) the lack of novelty (It seems that the proposed method simply combines existing neural networks and applies it to multi-agent behavior prediction.). There’s a huge recent development on multi-agent RL and IL regarding the scalability of MARL (MF-MARL, https://arxiv.org/pdf/1802.05438.pdf), coordinated multi-agent imitation learning (https://arxiv.org/abs/1703.03121), multi-agent GAIL (https://arxiv.org/abs/1807.09936), etc, which should be considered as related literature. In addition to them, there’s a paper on arXiv that uses GNN for MARL (https://arxiv.org/abs/1810.09202), which may be deeply related to this work as well.

I’ll definitely increase my score if I underestimate the quality of this work.

**Experience Assessment:**

I have published one or two papers in this area.

**Review Assessment: Checking Correctness Of Derivations And Theory:**

N/A

**Review Assessment: Checking Correctness Of Experiments:**

I carefully checked the experiments.

**Review Assessment: Thoroughness In Paper Reading:**

I made a quick assessment of this paper.

---

> ### Author Response · Authors · 2019-11-15
> **Responses**
>
> We thank the reviewer's input on alternative works, but argue with respect that the comparison between our work and RL based approaches does not apply.
>
> Our paper introduces a working model using standard ideas of GNN. Without claiming the superiority of our model to all other approaches, we establish the credibility of our model through comparison with a few alternative models and graphical demonstration of our model's prediction. Using the working model, we discuss the main contribution of our paper, and dedicate half of its length to the analysis of the scalability issue of the whole class of GNNs defined with these standard equations.

---

### Official Review · AnonReviewer4 · 2019-10-23
**Official Blind Review #4**

**Rating:** 3

**Review:**

The authors have proposed a GNN implementation that predicts and imitates the motion behaviors from observed swarm trajectory data. I have the following major comments on the paper:
1. I think the authors should discuss more on the related works. They should clearly mention their contributions and difference compared to the prior works.
2. What do the authors mean by 'functions universal across all nodes and edges'? Are these same functions for all nodes and edges?
3. I feel that the section 2.3 explaining 1D convolutions is slightly separated from the previous and following sections.
4. Why does the proposed GNN model performs better than Kipf's GNN for predicting and imitating the motion behaviors? What is the component that makes the difference?
5. How does the model depend on the history window length T_w? Also I am curious how does the model depend on the number of GNN layers used?
6. I am not quite satisfied with provided experimental evaluation of the paper. It is not clear to me why Kipf's GNN does not perform better than the proposed model (which simply aggregates the edge states). Also it would be interesting to try some other GNN models, such as ClusterGCN, GCI etc.
7. Also I want to see a fair comparison in the paper. If the main contribution is a GNN model, then the model should be compared with the recent GNN models, and if the main contribution is predicting the motion behavior from observed swarm trajectory data, it should be compared with MAS based approaches. While reading the paper, it appeared to me that the authors proposed a graph-based method for solving MAS task and they compared with an old GNN method, which is not justified.
I also have some minor comments as follows:
1. In page 1, "re-finement" should be "refinement"
2. In page 3, "... my be found in Battaglia ..." my should be may.
3. In page 4, "... sample rate is affects ..." should be "... sample rate affects ..."
4. In page 6, "... a agent ..." should be "... an agent ..."
5. In page 7, "... swarms of with larger ..." either of or with
6. In page 8, Table 3 ending parenthesis is missing.

**Experience Assessment:**

I have published one or two papers in this area.

**Review Assessment: Checking Correctness Of Derivations And Theory:**

I assessed the sensibility of the derivations and theory.

**Review Assessment: Checking Correctness Of Experiments:**

I assessed the sensibility of the experiments.

**Review Assessment: Thoroughness In Paper Reading:**

I read the paper at least twice and used my best judgement in assessing the paper.

---

> ### Author Response · Authors · 2019-11-15
> **Responses to Comments 2-7**
>
> 2. When we introduced the functions in a GNN, by "functions universal to all nodes and edges", we mean that the set of functions on each node's neighborhood is the same for all nodes. The function that computes pair-wise interactions between two nodes through an edge is also the same for all edges.
>
> 4. Our model outperforms Kipf's GNN model largely because of its lighter weight architecture. Kipf's model attempts to first infer the types of each edge through an encoder before assigning a different edge function for the decoder. Edge type inference is prone to failure in our experience with Kipf's model. Our model is supplied with the connectivity matrix for the tasks at hand, so the edge types (whether there is a connection or not) is known to the network, as is shown in the algorithm block. We also use different layer structures for each function in GNN, whereas Kipf elected to share the same MLP structure for all three functions, which makes it harder to tune.
>
> 5. For Markov processes, prediction should not depend on earlier history states, so $T_w$ greater than 1 should not improve the performance in principle. As far as the experiments are concerned, all three data sets are first-order deterministic Markov process, so the choice of $T_w$ is trivial in the scope of this paper. However, model is also applicable to higher-order Markov processes, in which the dependency on $T_w$ is an interesting topic to explore.
>
> 6 & 7. We admit the model comparison is a weak part of the paper. We argue that the main focus and contribution is the scalability of GNN, using motion prediction of multi-agent motion as a ground for discussion. To our best knowledge at the time of submission, no paper has concretely addressed the scalability issue of GNN, or they have simply glanced over the natural scalability of GNN noting the locality of the functions. The model comparison part of paper is not intended as an evidence that our implementation of GNN is superior to all of its counterparts, but to show a working model against baselines. With a model that works substantially well, we then move on to analysing and tackling the difficulties in scalability.

---

### Decision · Program_Chairs · 2019-12-19

**Decision:**

Reject

**Comment:**

This paper proposes a graph neural network based approach for scaling up imitation learning (e.g., of swarm behaviors). Reviewers noted key limitations in the discussion of related work, size of the proposed contribution in terms of model novelty, and evaluation / comparison to strong baselines. Reviewers appreciated the author replies which resolved some concerns but agree that the paper is overall not ready for publication.